

# On the colour of noctilucent clouds

Anna Lange [1], Gerd Baumgarten [2], Alexei Rozanov [3], and Christian von Savigny [1]

[1]Institute of Physics, University of Greifswald, Felix-Hausdorff-Str. 6, 17487 Greifswald, Germany
[2]Leibniz-Institute of Atmospheric Physics at the University of Rostock, Schlossstraße 6, 18225 Kühlungsborn, Germany
[3]Institute of Environmental Physics, University of Bremen, Otto-Hahn-Allee 1, 27359 Bremen, Germany

**Correspondence:** Anna Lange (s-anlang@uni-greifswald.de)

**Abstract.** The high-latitude phenomenon of noctilucent clouds (NLCs) is characterised by a silvery-blue or pale blue colour. In this study, we employ the radiative transfer model SCIATRAN to simulate spectra of solar radiation scattered by NLCs for a ground-based observer and assuming spherical NLC particles. To determine the resulting colours of NLCs in an objective way, the CIE (International Commission on Illumination) colour matching functions and chromaticity values are used. Different

processes and parameters potentially affecting the colour of NLCs are investigated, i.e., the size of the NLC particles, the abundance of middle atmospheric $O_3$ and the importance of multiply scattered solar radiation. We affirm previous research indicating that solar radiation absorption in the $O_3$ Chappuis bands can have a significant effect on the colour of the NLCs. A new result of this study is that for sufficiently large NLC optical depths and for specific viewing geometries, $O_3$ plays only a minor role for the blueish colour of NLCs. The simulations also show that the size of the NLC particles affects the colour of

the clouds. Cloud particles of unrealistically large sizes can lead to a reddish colour. Furthermore, the simulations show that the contribution of multiple scattering to the total scattering is only of minor importance, providing additional justification for the earlier studies on this topic, which were all based on the single scattering approximation.

## 1   Introduction

Noctilucent clouds (NLCs), also known as polar mesospheric clouds (PMCs), occur at latitudes poleward of about 50° in

the summer hemisphere at altitudes between about 80 and 85 km, slightly below the high latitude summer mesopause (e.g. Rapp and Thomas, 2006). The low temperature and a sufficient amount of water vapour at the summer mesopause lead to the formation of optically thin ice clouds (e.g. Gadsden and Schröder, 1989; Thomas et al, 1995; Baumgarten and Fiedler, 2008; von Savigny et al., 2020). NLCs were first reported by Backhouse (1885) and Leslie (1885) in 1885, two years after the Krakatoa volcanic eruption in 1883.



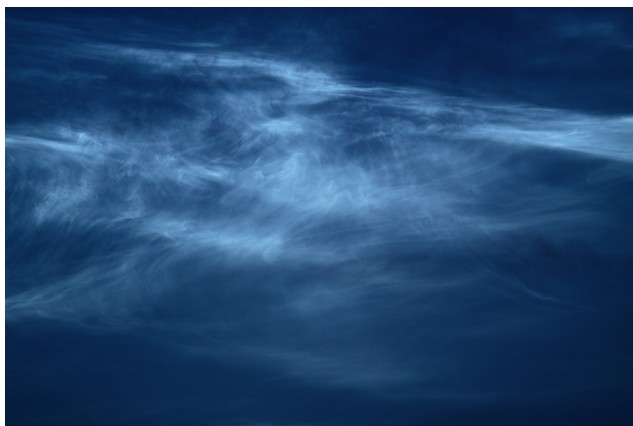

**Figure 1.** Photograph of NLCs taken by Gerd Baumgarten on July 14, 2009 from Djurhamn, Schweden.

Because they are very tenuous clouds, with vertical optical depths of typically $< 10^{-4}$ (Debrestian et al., 1997), they can only be seen during twilight when the Sun is between $6°$ and $16°$ below the horizon, and the clouds are still sunlit, while the observer and the atmosphere below the clouds are in darkness (Avaste et al., 1980; Thomas, 1991). NLCs appear "silvery", "pearly" and generally have a blue tint (Currie, 1962; Paton, 1964; Fogle, 1966). Figure 1 shows a typical example of an NLC with a blueish colour. Typical particle radii of visible NLCs are about $10 - 80\,\mathrm{nm}$ (e.g. Gumbel and Witt, 1998; von Savigny

et al., 2005; Baumgarten and Fiedler, 2008; Robert et al., 2009), so they are smaller than the wavelength of visible radiation and therefore preferentially scatter the short-wave blue light. Selective absorption of longer wavelengths in the Chappuis bands of ozone is also important for the colour of NLCs (Gadsden, 1975). Hulburt's 1953 paper demonstrated that ozone absorption must be taken into account for the correct modelling of colours and spectra of the twilight sky, including the zenith (Hulburt, 1953). Studies on the colours of NLCs are also found in the papers of Gadsden (1975) and Ostdiek and Thomas (1993).

Gadsden analysed spectral radiance and small-field spectral polarisation measurements and highlighted ozone as a decisive factor influencing the colour of NLCs (Gadsden, 1975). In contrast, Ostdiek and Thomas investigated the influence of two different cloud particle size distributions on the colour (chromaticity values) of NLCs (Ostdiek and Thomas, 1993). In the current work, the impact of several parameters is investigated that potentially influence the colour of NLCs. This includes the effect of ozone, the NLC particle size and the contribution of multiply scattered solar radiation. For this study we use

the radiative transfer model SCIATRAN developed by the Institute of Environmental Physics at the University of Bremen, Germany (Rozanov et al., 2014). Furthermore, the colours corresponding to the calculated spectra are determined and displayed using a standard approach, based on the CIE XYZ colour system (Wyszecki and Stiles, 2000; CIE, 2004) and the sRGB colour space.

    The paper is structured as follows. In Sect. 2 we introduce the main features of the SCIATRAN radiative transfer model

relevant to this study, as well as the colour modelling approach employed here. Section 3 presents the main results, i.e. the dependence of the colour of NLCs on the abundance of stratospheric ozone, on the NLC particle size and other parameters. The main implications and limitations of the results are discussed in Sect. 4 and conclusions are presented in Sect. 5.



## 2 Methodology

### 2.1 Radiative transfer simulations: SCIATRAN with incorporated Mie Code

To model the sunlight scattered by NLC particles and transmitted to the Earth's surface, the Mie Code implemented into the radiative transfer software SCIATRAN was used. This allows the calculation of aerosol optical parameters by SCIATRAN and the simultaneous implementation as an aerosol layer at a certain height. The NLC particle size distribution was assumed to be mono-modal log-normal:

$$n(r) = \frac{N_0}{\sqrt{2\pi} \cdot \ln(S) \cdot r} \cdot \exp\left[-\frac{(\ln r - \ln r_m)^2}{2\ln^2(S)}\right], \tag{1}$$

where $N_0$ is the total particle number density, $r_m$ the median radius, $r$ the particle radius and $S$ the geometric standard deviation of the distribution (Grainger, 2017). The calculations were carried out for median radii ranging from 10 to 1000 nm and constant values for $S = 1.4$ and $N_0 = 100$ cm$^{-3}$. Note that the vertical optical depth of the cloud layer is additionally specified, which leads to an adjustment of the value of $N_0$. The input values were guided by previous studies and literature on this topic (e.g., Gadsden and Schröder, 1989; Baumgarten and Fiedler, 2008; Baumgarten et al., 2010). In order to simulate the

solar radiation scattered by aerosols and air molecules in a spherical atmosphere, considering refraction effects for the direct solar beam and the scattered light, the "spher_scat" mode was used in SCIATRAN (Rozanov et al., 2014). SCIATRAN was developed by the Institute of Environmental Physics (IUP) of the University of Bremen as a forward model for the retrieval of atmospheric parameters from measurements with the SCIAMACHY instrument on ESA's Envisat spacecraft. More information on SCIATRAN can be found at https://www.iup.uni-bremen.de/sciatran/ (last access: March 7, 2022). The model output

contains radiance values at different wavelengths. These data were multiplied by the solar spectrum incident on the Earth's atmosphere (SORCE data (Solar Radiation and Climate Experiment)) (LASP, 2003) to obtain the resulting spectral distribution of the solar radiation scattered to an observer at the Earth's surface.

### 2.2 Colour modelling

The colour corresponding to a given scattered solar spectrum was determined and displayed using a standard approach based

on the CIE XYZ colour space established in 1931 (e.g., Wyszecki and Stiles, 2000; CIE, 2004; Brainard and Stockman, 2010). Using the CIE colour matching functions $\overline{x}(\lambda)$, $\overline{y}(\lambda)$ and $\overline{z}(\lambda)$ after Judd (1951) and Vos (1978), which quantify the spectral sensitivity of the three cone cells of the human eye, the CIE tristimulus values $X$, $Y$ and $Z$ are determined (Billmeyer Jr. and Fairman, 1987):

$$X = k \int\limits_{380\,\text{nm}}^{800\,\text{nm}} I(\lambda)\,\overline{x}(\lambda)\,d\lambda \tag{2}$$


$$Y = k \int\limits_{380\,\text{nm}}^{800\,\text{nm}} I(\lambda)\,\overline{y}(\lambda)\,d\lambda \tag{3}$$





$$Z = k \int\limits_{380\,\text{nm}}^{800\,\text{nm}} I(\lambda)\,\overline{z}(\lambda)\,d\lambda, \tag{4}$$

where $I(\lambda)$ is the given radiance spectrum and the normalizing factor k is defined as

$$k = \frac{100}{\int_{380\,\text{nm}}^{800\,\text{nm}} I_{\text{achromatic}}(\lambda)\,\overline{y}(\lambda)\,d\lambda}, \tag{5}$$

with $I_{\text{achromatic}}(\lambda)$ as a reference spectrum with the colour impression of white. In the case of self-luminaries, $k$ remains indeterminate. Based on the XYZ tristimulus values the CIE chromaticity values $x$ and $y$ are calculated using

$$x = \frac{X}{X+Y+Z} \qquad y = \frac{Y}{X+Y+Z}. \tag{6}$$

These chromaticity values characterize the colour independently of the brightness and are displayed in a 2-D plot, the so-
called CIE chromaticity diagram or "Gamut". Furthermore, the XYZ tristimulus values were converted to sRGB (standard RGB), which can be used to display the colours in the programming software IDL (Interactive Data Language). More detailed information can be found in a previous paper by Wullenweber et al. (2021).

## 3 Results

For the radiative transfer simulations carried out in this work, SCIATRAN version 4.1.3 (Rozanov et al., 2014) is used. Standard
atmospheric trace gas profiles (including $H_2O$, $O_3$, $O_2$, $CO_2$, $SO_2$, $NO_3$ and $N_2O$), as well as pressure and temperature profiles for high mid-latitudes taken from a climatological database obtained from a 3-D CTM (chemical transport model) developed at the University of Bremen (Sinnhuber et al., 2003) are used. In addition, the "DOM_S" (scalar computation) setting is used, which means that the radiative transfer equation is solved with a scalar discrete ordinate approach (Rozanov et al., 2014) and with "the number of iterations" = 1, an approximate treatment of multiple scattering is performed. This mode is referred to as
the approximate spherical solution. The errors resulting from this simplified treatment are small, as further discussed in Sect. 3.5. Figure 2 illustrates the viewing geometry in SCIATRAN, which is essentially defined by three angles: the solar zenith angle (SZA), the solar azimuth angle (SAA), and the viewing zenith angle (VZA) in [deg]. The viewing zenith angle defines the line-of-sight angle at the observer position with a maximum value of $90°$ for a ground-based observer. That means at a VZA of $0°$ the imaginary observer looks to the zenith and at $90°$ to the horizon. The solar azimuth angle describes the azimuth angle
of the Sun's position with respect to the viewing direction. The value of $0°$ corresponds to the solar direction, and the value of $180°$ to the anti-solar direction. At this point it should be noted that due to the azimuthal symmetry in SCIATRAN, the values between $180°$ and $360°$ describe the same viewing geometry as the corresponding values between $180°$ and $0°$ (Rozanov et al., 2014). With these three angles it is therefore possible to specify the geometry based on the position of the Sun and the observer.




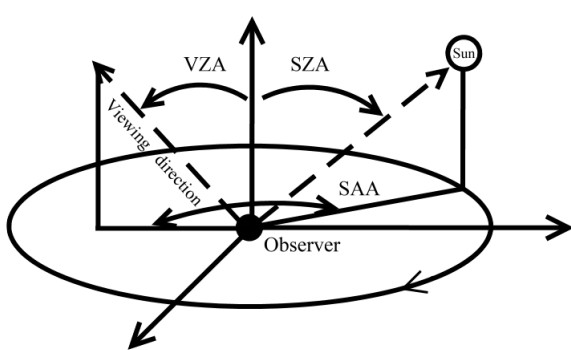

**Figure 2.** Definition of the viewing geometry in SCIATRAN. With: SZA (solar zenith angle), VZA (viewing zenith angle) and SAA (solar azimuth angle).

## 3.1 Impact of the NLC optical depth

Figure 3 shows scattered solar spectra determined by multiplying the SORCE solar spectrum (LASP, 2003) by the scattered radiance spectra simulated with SCIATRAN. The left panel of Fig. 3 includes an NLC with the following characteristics: $r_m$ = 50 nm, $S$ = 1.4, vertical optical depth of $\tau_{\mathrm{NLC}} = 10^{-4}$, 1 km vertical extent and a center altitude of $z_{\mathrm{NLC}}$ = 82 km. The right panel shows the background spectrum without NLCs. Both spectra correspond to a solar zenith angle (SZA) of 98°, a solar azimuth angle (SAA) of 0°, and a viewing zenith angle (VZA) of 65°. Note that the radiances in the case with NLCs are about

an order of magnitude larger than in the background case.





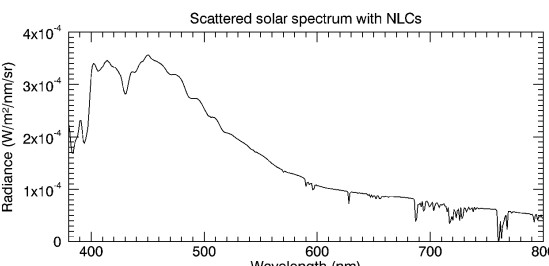
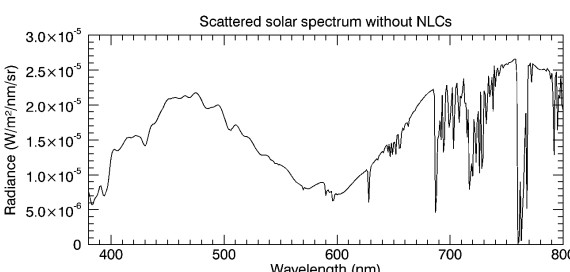

**Figure 3.** Solar scattering spectra at the Earth's surface. Including NLCs with an optical depth of $10^{-4}$ at an altitude of $82\,\mathrm{km}$ (left panel) and without NLCs (right panel), simulated for a solar zenith angle of $98°$ and a viewing zenith angle of $65°$ in the solar direction.

At twilight – i.e. the sun being below the horizon – NLCs appear to an observer on the Earth's surface with a bluish colour. This is determined by the absorption of solar radiation in the $O_3$ Chappuis bands with maxima at $575\,\mathrm{nm}$ and $603\,\mathrm{nm}$, filtering the longer wavelengths and the NLC particle size distribution parameters (here: $r_m = 50\,\mathrm{nm}$ and $S = 1.4$). So the scattered solar radiation including NLCs appears blue (left panel of Fig. 3). Without NLCs (right panel of Fig. 3) the spectrum also exhibits a peak at about $750\,\mathrm{nm}$, resulting in a slightly different blue hue (Fig. 4). Figure 4 shows a CIE chromaticity diagram with the chromaticity values x and y on the axes. The arc with the filled colour circles represents the positions of the spectral colours with the corresponding wavelengths in the x-y plane. The connecting line at the bottom of the arc cannot be represented by pure spectral colours and is called the "line of purples". The colours in the diagram are based on the conversion of the chromaticity values to sRGB as described in Sect. 2.2. The small "x" corresponds to the chromaticity values of the unattenuated solar spectrum. Simulations for other optical depths (in the range of $10^{-3}$ to $10^{-6}$) show only minor differences in the resulting colours, which is why for the sake of clarity a separate presentation is omitted.

Figures 5 and 6 show solar scattering spectra (left column) with the resulting colour impression (right column) for an observer on the Earth's surface. The simulations were carried out for SZA = $98°$, viewing zenith angles of $10°$, $20°$, $40°$, $60°$, $80°$ (from top to bottom) and a solar azimuth angle of $0°$. The simulations differ in the assumed vertical optical depth of the NLCs: $\tau_{\mathrm{NLC}} = 10^{-4}$ (Fig. 5) and $\tau_{\mathrm{NLC}} = 10^{-5}$ (Fig. 6). Both sets of plots show the Chappuis bands of ozone, whose visibility decreases with increasing VZA (from top to bottom). Furthermore, these simulations also show that the Chappuis bands are more clearly visible with decreasing NLC optical depth (compare Figs. 3, 4, 5 and 6) and the spectral maximum at about $750\,\mathrm{nm}$ has no noticeable effect on the colour of NLCs. Considering that single scattering is a valid approximation as discussed in Sect. 3.5, these observations can be explained by following geometrical considerations: With NLCs at an altitude of $82\,\mathrm{km}$, the scattered radiation comes primarily from this altitude and is hardly affected by stratospheric ozone due to the slight atmospheric penetration of solar radiation on its path to the NLC, at least for the SZA considered here. For SZA = $98°$ and VZA = $65°$, the tangent height of the solar beam is about $40\,\mathrm{km}$. Without NLCs the scattered light is purely Rayleigh



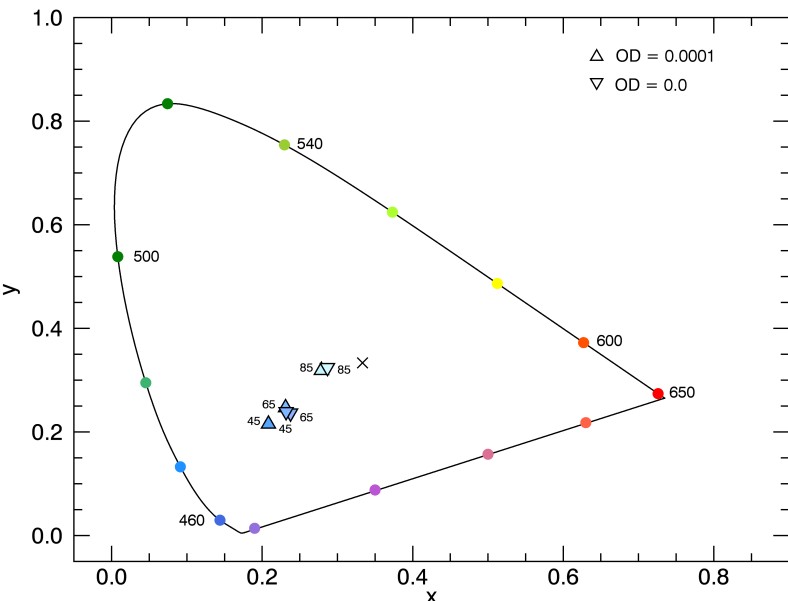

**Figure 4.** CIE chromaticity diagram corresponding to Fig. 3 with NLCs (OD = $10^{-4}$) and without NLCs (OD = 0). Marked at the data points: VZAs = $45°$, $65°$ and $85°$, with SZA = $98°$ and SAA = $0°$.

scattered and since the density in the atmosphere increases exponentially with decreasing altitude, the scattered radiation comes from lower altitudes and is more affected by the stratospheric ozone. Accordingly, the $O_3$ Chappuis bands are more clearly visible. Therefore, with a sufficiently large NLC optical depth and a certain viewing geometry, the NLC signal dominate over the background signal and the Chappuis absorption is no longer visible. However, the NLCs still appear blueish. The finding that absorption in the Chappuis bands of $O_3$ is not required to explain the blue colour of NLCs in some situations is a new result compared to earlier works (e.g. Gadsden, 1975).




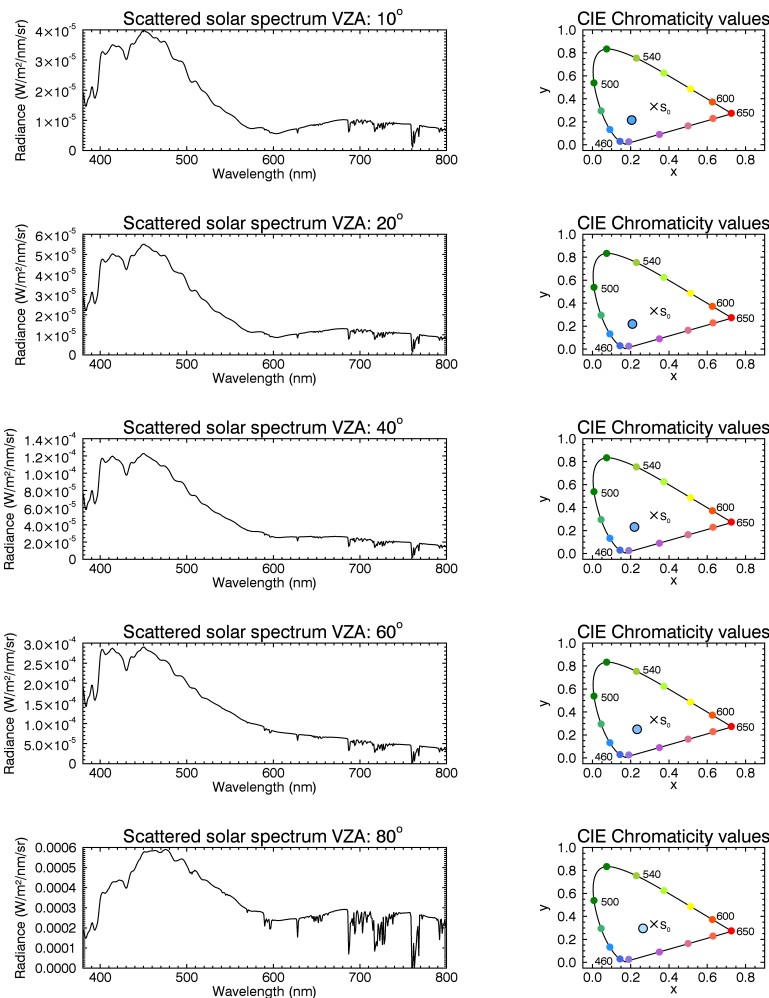

**Figure 5.** Solar scattering spectra (left column) and CIE chromaticity diagrams (right column) for an observer on the Earth's surface and for a SZA of 98° and VZAs of 10°, 20°, 40°, 60°, 80° (from top to bottom) and a SAA of 0°. Including NLCs with an optical depth of $10^{-4}$ at an altitude of 82 km.

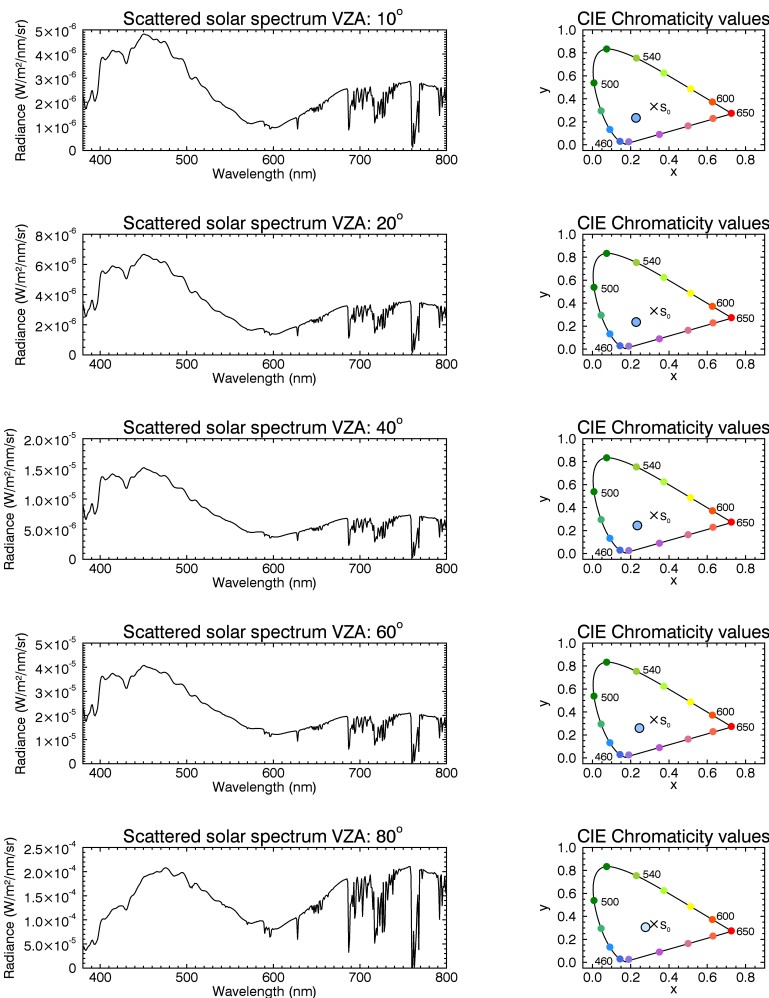

**Figure 6.** Solar scattering spectra (left column) and CIE chromaticity diagrams (right column) for an observer on the Earth's surface and for a SZA of $98°$ and VZAs of $10°$, $20°$, $40°$, $60°$, $80°$ (from top to bottom) and a SAA of $0°$. Including NLCs with an optical depth of $10^{-5}$ at an altitude of $82\,\mathrm{km}$.





## 3.2 Impact of ozone absorption

As evident from the spectra in Fig. 3, ozone absorption may also affect the colour of noctilucent clouds. The effect of ozone was already investigated by Gadsden (1975), who made measurements of the spectral radiance of NLCs with a photoelectric spectropolarimeter. Figure 7 shows a CIE chromaticity diagram including NLCs and different ozone column densities.

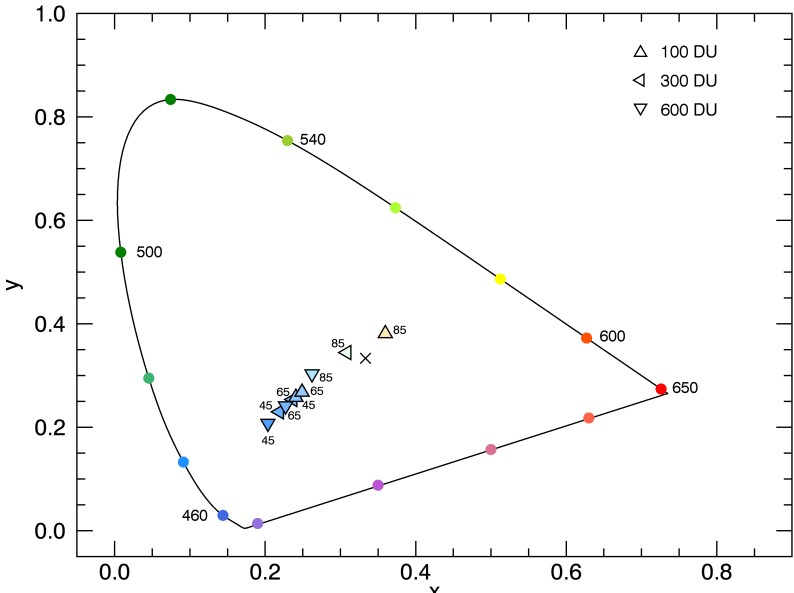

**Figure 7.** CIE chromaticity diagram for spectra including NLCs and different ozone column densities. Marked at the data points: VZAs = $45°$, $65°$ and $85°$, with SZA = $98°$ and SAA = $0°$. The NLC parameters are as for the left panel of Fig. 3, i.e.: $r_m$ = $50\,\text{nm}$, $S$ = $1.4$, $\tau_{\text{NLC}}$ = $10^{-4}$ and $z_{\text{NLC}}$ = $82\,\text{km}$.

For a vertical ozone column density of 300 DU, the colour changes with the VZAs ($45°$, $65°$ and $85°$) from dark blue to light blue. This corresponds to a natural colour gradient of NLCs during twilight. With more ozone (600 DU), a shift of colours

to smaller x values, i.e. a blue shift, can be observed. With a lower ozone column density (100 DU) a shift to larger x-values occurs. This can be explained by the effect of ozone absorption in the Chappuis bands. Due to the long light path through the atmosphere, green, yellow, orange and short-wave red light is effectively absorbed by ozone, so that blue light predominates. With more ozone, this effect is intensified and leads to a more saturated blue colour (compare 600 DU). However, a noticeable colour change only occurs at a low ozone column density (100 DU). This is due to the lower attenuation of the long-wave light

by ozone absorption, resulting in a shift to the reddish region of the Gamut (see VZA = $85°$). In addition, the effect of ozone absorption increases for larger VZAs, due to the longer path through the ozone layer from the observer to the NLC. Since the





changes in colour and the positions in the CIE chromaticity diagram corresponding to the different ozone column densities are close together, it requires an unrealistically small amount of ozone for the colour of the NLCs to change. Nevertheless, ozone absorption must be taken into account to explain the colour of noctilucent clouds.

## 3.3  The role of particle size

The typical particle radii of visible NLCs are in the range of 10 – 80 nm (see Sect. 1). In order to test the effect of the NLC particle size on the colour of the clouds, we performed SCIATRAN simulations for different median radii of the assumed mono-modal log-normal particle size distribution, i.e. 10 nm, 50 nm, 200 nm, 600 nm and 1000 nm. The width parameter is kept constant at $S = 1.4$ and the optical depth was assumed to be $\tau_{\mathrm{NLC}} = 10^{-4}$ in all cases. Figure 8 shows a chromaticity

diagram with simulated colours for increasing particle sizes.

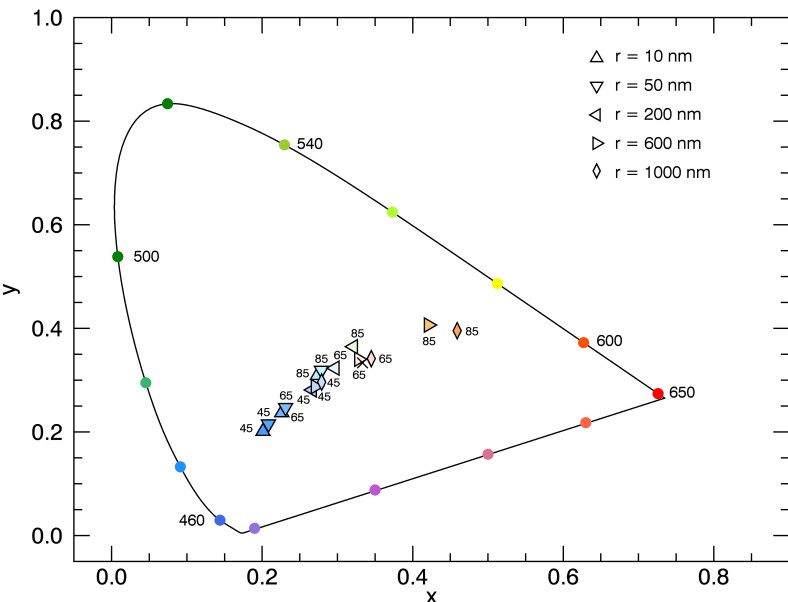

**Figure 8.** CIE chromaticity diagram for simulations with NLCs and different median radii of the NLC particle size distribution. Marked at the data points: VZAs = 45°, 65° and 85°, with SZA = 98° and SAA = 0°. The width parameter of the size distribution is S = 1.4 and the optical depth is $\tau_{\mathrm{NLC}} = 10^{-4}$.

As expected, solar radiation scattered by particles with typical radii (10 to 50 nm) is perceived as blue by a ground-based observer. In addition, these radii also show the colour change from dark blue to white blue / light blue with the VZAs (45°, 65° and 85°). Assuming significantly larger particles, the scattered light becomes more reddish (compare 600 and 1000 nm). That means, when the particles become larger and scatter spectrally more neutral, the reddish colouring is also visible from the





Earth's surface. Most of the calculations by Ostdiek and Thomas (1993), who have summarised various NLC measurements, are in agreement with our results, only in one case their calculations show smaller particles in the yellowish/orange region of the CIE chromaticity diagram. The simulations displayed in Fig. 8 show that the typical colour of NLCs is only present for certain particle sizes. Furthermore, it can be shown that the light scattered by NLCs in the visible spectral range contains important information on the size of NLC particles.

**3.4 Influence near the horizon**

During sunset, a reddening appears on the horizon (see Fig. 9), which accompanies most NLC observations (depending on the SZA). Figures 10 and 11 show simulated solar scattering spectra (left column) with the resulting colour impression (right column) for a ground-based observer and SZA = 98°, VZA = 84°, 87°, 90° (from top to bottom) and SAA = 0°. Figure 10 shows the calculated results for NLCs with the following parameters: $r_m$ = 50 nm, $S$ = 1.4, vertical optical depth of $\tau_{\mathrm{NLC}}$ =

$10^{-4}$, and an altitude of $z_{\mathrm{NLC}}$ = 82 km. In comparison, Fig. 11 illustrates the background without NLCs.

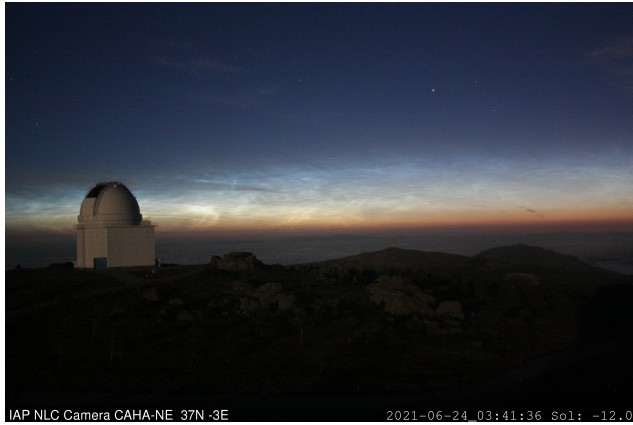

**Figure 9.** Photograph of NLCs taken by Gerd Baumgarten on June 24, 2021 from Calar Alto, Spain.

Both plots depict the colour change from blue to orange near the horizon. Note here that the radiance values of the maximum in the short-wave blue spectral range at about 470 nm are larger for the simulations with NLCs than for the background case, especially for VZA = 84° (upper panel). This results in different positions in the CIE chromaticity diagram. In contrast, the spectra for VZA = 90° (lower panel) show no significant differences. Due to the very small radiance values for the case with

NLCs in the range of about 400 nm, the positions in the CIE chromaticity diagram differ, but not the resulting colour impression of orange. Since the simulated spectra near the horizon barely deviate in intensity and spectral shape, it can be concluded, as expected, that NLCs play no decisive role in the red colouring of the horizon.





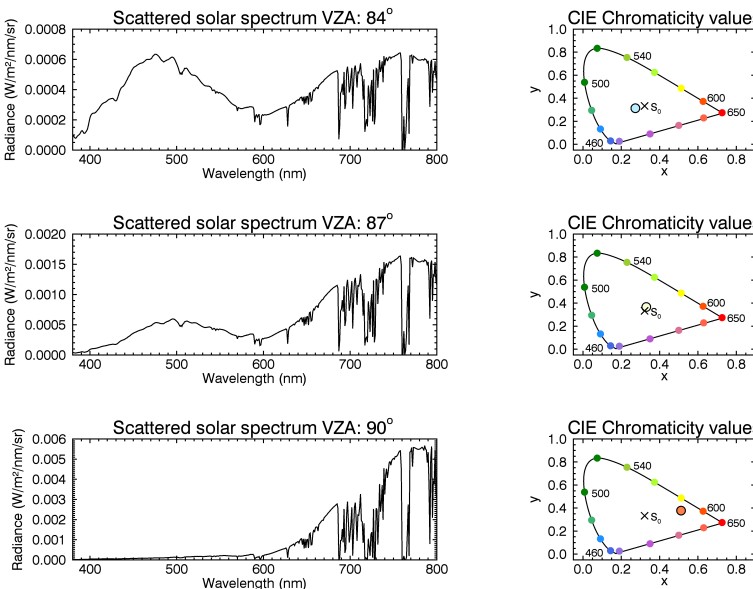

**Figure 10.** Solar scattering spectra (left column) and CIE chromaticity diagram (right column) for an observer on the Earth's surface and for SZA = 98°, VZA = 84°, 87°, 90° (from top to bottom) and SAA = 0°. The simulations are calculated for NLCs with following parameters: $r_m$ = 50 nm, $S$ = 1.4, $\tau_{\mathrm{NLC}} = 10^{-4}$ and $z_{\mathrm{NLC}}$ = 82 km.

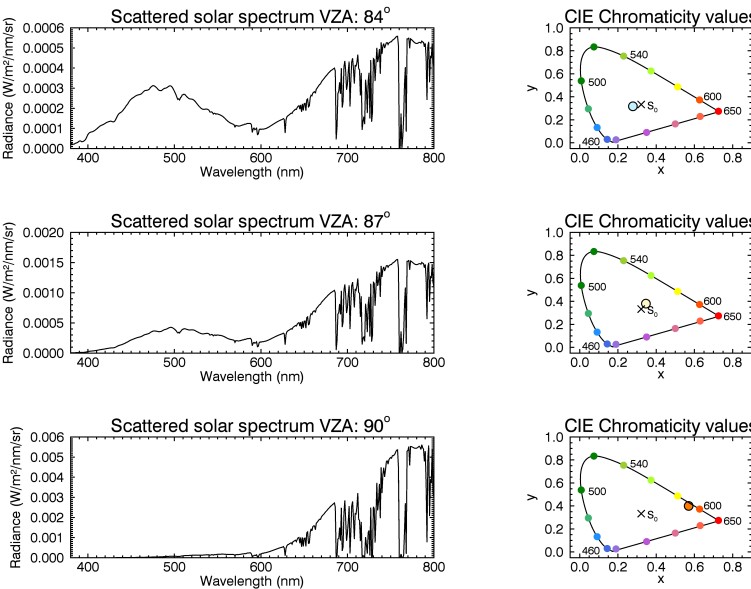

**Figure 11.** Solar scattering spectra (left column) and CIE chromaticity diagram (right column) for an observer on the Earth's surface and for SZA = 98°, VZA = 84°, 87°, 90° (from top to bottom) and SAA = 0°. The calculated spectra show the background without NLCs.





### 3.5 Multiple scattering vs. single scattering

In earlier studies on simulations of the spectral distribution of solar radiation scattered by NLC particles, the contribution
of multiply-scattered radiation has been neglected (e.g. Gadsden, 1975; Ostdiek and Thomas, 1993). Using SCIATRAN, the
contribution of multiple scattering to the NLC spectra as seen by a ground-based observer can be easily simulated. For calcula-
tions with a more accurate consideration of multiple scattering, which is referred to as the fully spherical solution, i.e. "number
of iterations" $> 1$ (see Sect. 3), SCIATRAN version 4.5.5 is now used. Figure 12 shows the difference of both methods for
scattered solar spectra with VZA = 65° (left panel) and the ratio for different VZAs (right panel).

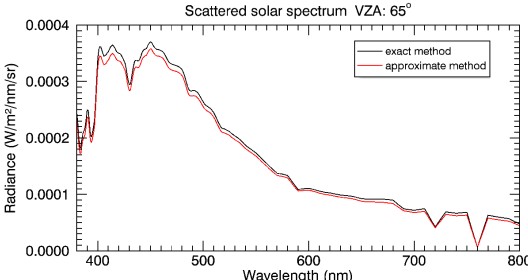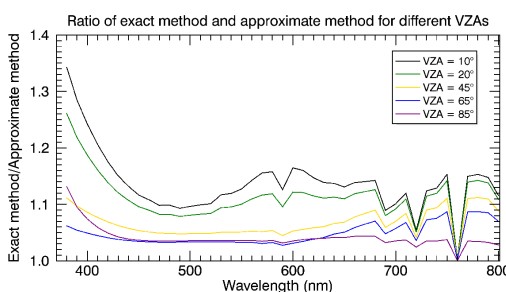

**Figure 12.** Left panel: Solar scattering spectra for VZA = 65° calculated with the exact method (black line) and the approximate method
(red line). Right panel: Ratio of the exact method and the approximate method for different VZAs. For all simulations, SZA = 98° and SAA
= 0°. The NLC parameters are: $r_m$ = 50 nm, $S$ = 1.4, $\tau_{\mathrm{NLC}} = 10^{-4}$ and $z_{\mathrm{NLC}}$ = 82 km.

The differences are mainly in the short-wave blue spectral range (maximum factor of 1.34). Overall they are not crucial in
the context of the current study. This is especially the case for the NLC viewing geometry relevant VZAs here (45°, 65° and
85°). Furthermore, Fig. 13 shows that the more accurate consideration of multiple scattering has no visible effect on the CIE
chromaticity values and the resulting colour, which is due to the blue CIE colour matching function $\overline{z}(\lambda)$ having its maximum
at 450 nm. Therefore, the approximate multiple scattering treatment method is sufficient for the simulations performed here.



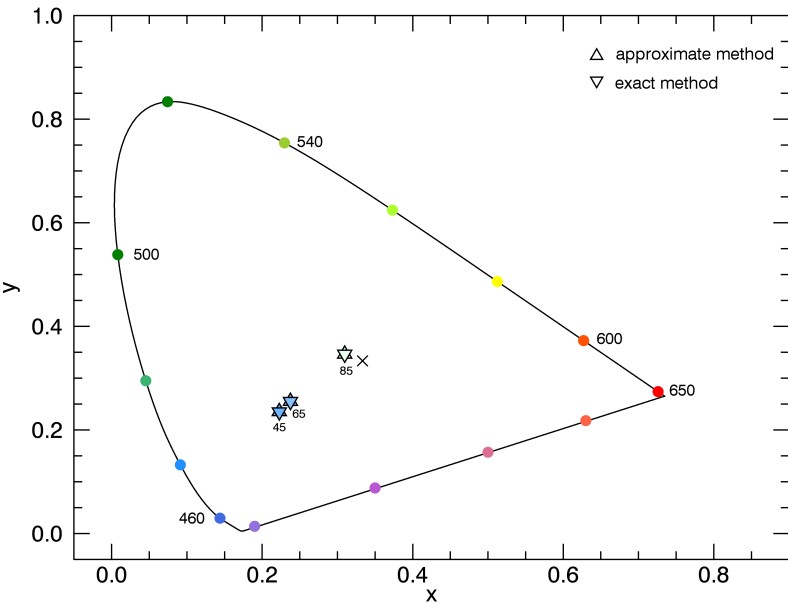

**Figure 13.** CIE chromaticity diagram for simulations with NLCs considering different methods for treating multiple scattering. Marked at the data points: VZAs = 45°, 65° and 85°, with SZA = 98° and SAA = 0°. The NLC parameters are: $r_m = 50\,\mathrm{nm}$, $S = 1.4$, $\tau_{\mathrm{NLC}} = 10^{-4}$ and $z_{\mathrm{NLC}} = 82\,\mathrm{km}$.

In comparison, Fig. 14 shows CIE chromaticity values for different viewing geometries (VZAs: 45°, 65°, 85° and SZA: 98°) and for single and multiple scattering simulations. For the calculations considering multiple scattering, the fully spherical solution was used here. The simulations show that for multiple scattering the colour is slightly bluer than for single scattering. This is due to Rayleigh scattering and the resulting preference for the short-wave blue light. However, no significant differences are observed.



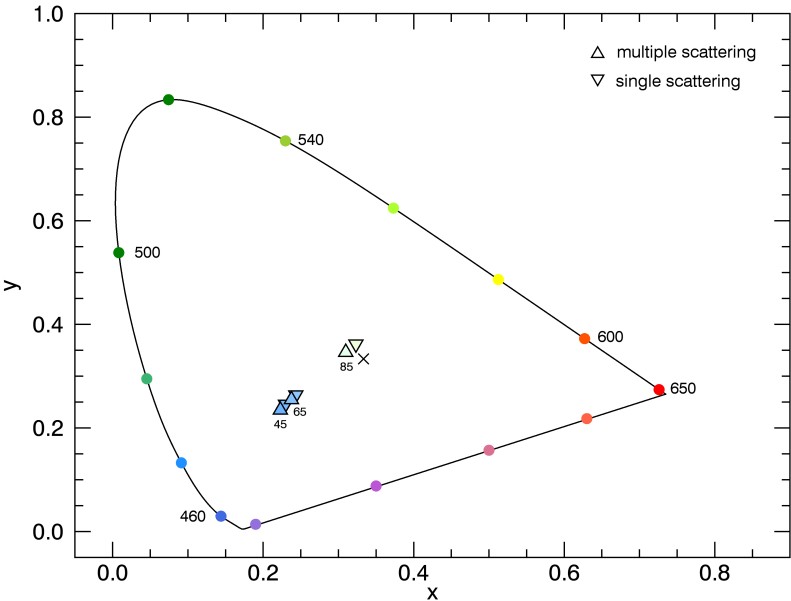

**Figure 14.** CIE chromaticity diagram for simulations with NLCs considering multiple scattering and single scattering only. Marked at the data points: VZAs = 45°, 65° and 85°, with SZA = 98° and SAA = 0°. The NLC parameters are: $r_m$ = 50 nm, $S$ = 1.4, $\tau_{\mathrm{NLC}}$ = $10^{-4}$ and $z_{\mathrm{NLC}}$ = 82 km.

The spectra in Fig. 15 (left panel) show the simulated spectral distribution with and without the multiple scattering contribution for VZA = 65°. The right panel compares the ratio of multiple scattering and single scattering for different VZAs.





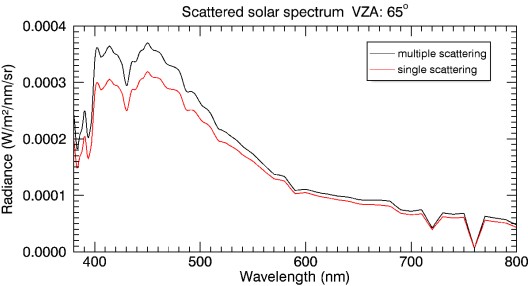
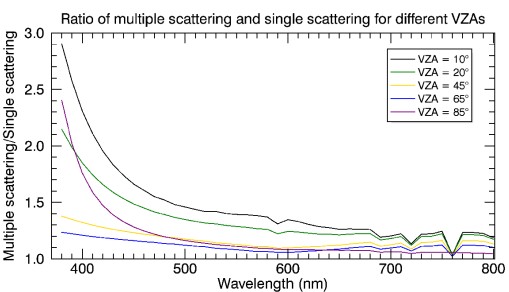

**Figure 15.** Solar scattering spectra for VZA = 65° with the multiple scattering (black line) and without the multiple scattering contribution to the total scattered radiance (red line). Right panel: Ratio of multiple scattering and single scattering for different VZAs. For all simulations, SZA = 98° and SAA = 0°. The NLC parameters are: $r_m = 50$ nm, $S = 1.4$, $\tau_{\mathrm{NLC}} = 10^{-4}$ and $z_{\mathrm{NLC}} = 82$ km.

The scattered solar spectra show that for the single scattering, the simulated values are smaller than for the multiple scattering (the maximum relative difference between the two spectra is 19%). Especially in the short-wave range the influence of multiple scattering is significant. Overall, the differences for the VZAs of the NLC viewing geometry used here are not significant with

respect to the focus of this study and confirm the results of Fig. 14. As above, the weak influence of the large differences at shorter wavelengths (compare Fig. 14) is due to the blue CIE colour matching function $\overline{z}(\lambda)$ with a maximum at 450 nm. However, the differences depend highly on the VZA. Near the zenith, multiple scattering has a large influence and results in a maximum factor of about 3.

Single scattering is a valid approximation for the NLC viewing geometry used here, but it should be noted that depending

on the relevant VZA range, the effect of multiple scattering may dominate and lead to different conclusions.

## 4  Discussion

We begin with a discussion of the limitations of the approach and results presented in this study, followed by a summary of how our results compare to the few earlier studies on the colour of NLCs.

Currently the study is limited to Solar elevation > -8°, which covers about 40% of all NLCs (Baumgarten et al., 2009, Fig.

3) due to the SZA limitation in SCIATRAN version 4.1.3. In the future we want to study non-spherical particles (Baumgarten et al., 2002; Hervig et al., 2009), however we do not expect a qualitative change of our results since previous studies have shown only little effect on colour ratios (Kiliani et al., 2015).

It also should be kept in mind that only the position of a given spectrum in the CIE chromaticity diagram provides objective information on the associated colour. The colours of the symbols displayed in the chromaticity diagrams depend on the details

of the calculation of the RGB values and will vary to a certain extent between different output devides.



In their work, Ostdiek and Thomas (1993) investigated the effect of two different NLC particle size distributions on the chromaticity values of NLC scattering spectra. The distributions were (1) a mono-modal log-normal distribution with an effective radius of 42.6 nm and (2) a power law distribution with an effective radius of about 700 nm. In good qualitative agreement with our results, the population of small particles leads to positions in the blue part of the chromaticity diagram, whereas the population of large particles leads to yellowish colours. Ostdiek and Thomas (1993) neglect refraction, Gadsden (1975) considers it, but argues that its effect is very small. Ostdiek and Thomas (1993) mention that a test was carried out that showed that refraction does affect the radiance values, but has a minor impact on the spectral shape of the scattering spectra and in subsequence also on the chromaticity values. Our simulations confirm the results of Ostdiek and Thomas (1993) (not shown). Gadsden (1975) also emphasized Chappuis absorption of ozone as a major factor influencing the colour of NLCs. However, our results show that for certain combinations of observation geometry and optical depth, ozone absorption is no longer visible and plays only a minor role for these cases.

## 5 Conclusions

In this work, various parameters that influence the colour of NLCs were investigated. The Mie theory was used for the calculations and therefore the assumption of spherical particles was made. To be able to make concrete conclusions about colour changes, the CIE chromaticity diagram was used.

First, an unrealistically small amount of ozone is required to observe a deviation from the typical blue colour of NLCs. A new result in this work is that for sufficiently large NLC optical depths and for specific viewing geometries, ozone plays only a minor role for the blueish colour of NLCs. Second, the particle size decisively determines the perceived colour of NLCs. From this it follows that the typical colour is only observable for certain particle sizes. Therefore, some information about the size of the particles can be derived from the colour of the scattered light in the visible spectral range. Third, NLCs do not influence the reddish colour of the horizon. Fourth, the difference between the single and the multiple scattering plays a negligible role for the perceived colour of NLCs geometries considered in this study.

*Code availability.* The SCIATRAN radiative transfer model can be accessed via the following link:
https://www.iup.uni-bremen.de/sciatran/ (last access: March 7, 2022).

*Author contributions.* AL and CvS outlined the project and AL carried out the SCIATRAN simulations with guidance by AR. GB provided NLC photographs and his expertise to NLCs. AL wrote an initial version of the paper. All authors discussed, edited and proofread the paper.

*Competing interests.* The authors declare that they have no competing interests.





*Acknowledgements.* The authors acknowledge financial support by the Deutsche Forschungsgemeinschaft and the University of Greifswald. This study was enabled by the collaborations within the DFG research unit Volimpact (FOR 2820, grant no. 398006378). We are indebted
to the Institute of Environmental Physics of the University of Bremen – particularly to Vladimir Rozanov and John P. Burrows FRS – for access to the SCIATRAN radiative transfer model. We are thankful for the generous support by Jens Helmling of Calar Alto Astronomical Observatory in Spain in operating the southernmost camera of our European camera network. The work benefitted from the support by Michael Priester and citizen scientists in detecting NLCs in our camera observations.



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
