# Peer review of "On the colour of noctilucent clouds"

_Annales Geophysicae, 2022_

## Author Comment (AC1)

**Replies to comments by reviewer 1**

**Comment**: In this work, the authors investigated the color of NLCs, quantitatively, based on Mie scattering process and updated NLC scattering simulation model. The main parameters to examine the scattered spectrum are particle size, ozone absorption and influence of multiple scattering. They found that the effect of ozone absorption was less in color (blueish) than the previously reported.

The process of calculation of the NLC color and presentation in Figure 4, 7, 8 are good for who are not familiar to treat the optical characteristics of NLCs.

My conclusion is that the present work is worth to publish with a minor correction.

**Reply**: We thank the reviewer for his/her constructive and helpful comments. We tried to answer every comment in an appropriate way.

**Comment**: Minor comments

Line 91-94, solar zenith angle and viewing angle: The authors did not comment on the refraction effect of the solar view angle. Please note how does it consider in their simulation ?

**Reply**: In our simulations performed with SCIATRAN, refraction effects are taken into account (as described in L. 55-56). L. 91-94 refer to the definition of the viewing geometry in SCIATRAN (Fig. 2). The aim of Fig. 2 was only to explain the different angles in SCIATRAN, which is why refraction effects are not part of it.

**Comment**: Lines 163-164 "it can be shown that the light scattered by NLCs in the visible spectral range contains important information on the size of NLC particles".: The authors mention that measuring the color of NLC they can estimate the size of NLC particles (Figure 8). If this is an important finding in the present work, they can show it in quantitatively (Color as a function of particle size with more detail).

**Reply**: "The authors mention that measuring the color of NLC they can estimate the size of NLC particles (Figure 8).": We did not say that. The colour also depends strongly on the OD (see Fig. 4). But it is the case that the particle size influences the colour and thus very large particles can be excluded. For a better understanding of this sentence (L. 163-164), we added additional explanations. It now says: "Furthermore, it can be shown that the light scattered by NLCs in the visible spectral range contains important information on the size of NLC particles. Thus, very large particles can be excluded by the resulting colour."

---

## Author Comment (AC2)

**Replies to comments by reviewer 2**

**Comment**: The paper describes the basic factors defining the visible colour of noctilucent clouds. These factors are listed in a paper, each of it is considered separately. The conclusions about importance of each factors are made. All required calculations are described, the results seem adequate and expected. However, I would suggest some corrections those can make the effects easier to be understood.

**Reply**: We thank the reviewer for his/her constructive and helpful comments. We tried to answer every comment in an appropriate way.

**Comment**: 1. Authors desribe the coordinate system in the paper, varying the Viewer zenith angle VZA and atmospheric parameters (ozone content by unrealistic factors up to 2-3, NLC optical depth and particle size up to huge values, models of multiple scattering, etc.). In the same time, the basic factor defining the NLC illumination, Solar zenith angle, SZA, remains constant, 98deg. In this case the most of colour variations caused by ozone Chappuis absorption, Rayleigh and aerosol extinction of tangent solar emission while the Sun sets beyond the limb remain unseen.

1a (continuation of previous). Conditions of NLC illumination by the Sun through the troposphere and stratosphere are defined not by SZA in observation point but by local $SZA_L$ visible from the cloud. In the same moment, NLC in different parts of the sky will have different local $SZA_L$ and - thus - different conditions of ilumination and different effects of ozone. This effect is in fact noticed by authors in Figures 5 and 6, where Chappuis bands become less noticeable at large VZA. This effect is defined by lower local $SZA_L$ there (about 97deg in these conditions), NLC are still illuminated by the rays propagated above the ozone layer. But if we take SZA in the observation point about 98.5-99.0deg, then the effects if ozone will be stronger for large VZA. In the same time the colour for small VZA will turn redder due to immersion of solar rays to the dense atmospheric layer with strong Rayleigh scattering.

Models and also the observations show that the ozone effect is maximal around local $SZA_L$ around 98deg. So, if NLC is seen near the zenith, the ozone effect will be strongest at the same observable SZA=98deg. But if we see NLC in the dusk area (SAA=0 and large VZA), then ozone bluening will reach the maximum later. Finally, I would suggest the authors considering this in details.

**Reply**: Thank you, we referred to the local solar zenith angle in Sect. 3.1 and described its use and conclusions in the discussion. We added the following points to the discussion:
- To study the conditions of NLC illumination by the Sun and the influence of ozone for different viewing geometries, the local solar zenith angle ($SZA_L$), i.e. the solar zenith angle at the location of the NLC can be used.
- NLCs in different parts of the sky have different $SZA_L$ resulting in different effects of ozone (in this work: each VZA corresponds to a different $SZA_L$)
- For the NLC viewing geometries used in this work, the effects and conclusions described in Sect. 3.1 remain unchanged with the consideration of the $SZA_L$. However, for different geometries and observations, the use of the $SZA_L$ can be helpful.

**Comment**: 2. Considering the problem of multiple scattering and its weak influence on NLC color, we can add that observations show the linear polarization of NLC field (p 1.0) at scattering angles around 90deg. This confirms the weak influence of multiple scattering or, at least, small angular size of secondary light source around the Sun. It is also the confirmation that single scattering models describe the changes of NLC color well. It can be added to the paper.

**Reply**: This is a good point. We thank the reviewer for pointing this out. We mentioned this now at the end of the discussion section and referred to a paper by Ugolnikov et al. (2016).

**Comment**: 3. Colour effects considered in the paper are quite small (about 0.1 in terms of values x and y), that is significantly less than scale of Figures 4, 7, 8, 13, 14. This makes these figures hard to analyze, especially Figures 7-8, where two parameters (VZA and ozone/particle size) vary. Maybe the figures should be rescaled or replaced by graphs with dependencies of colours x and y on each parameter separately.

**Reply**: Thank you for these suggestions. We added an enlarged figure of the relevant data points in the CIE chromaticity diagram for each of these figures (4, 7, 8, 13, 14).

**Comment**: 4. Effect of NLC and clear sky reddening near the horizon (chapter 3.4) is basically defined by extinction in the lowest atmospheric layers those conditions are unpredictable and can differ a lot from the model used in the paper. Moreover, at LZA around 90deg the Bouger's law can not be used. So, it should be added that it is only the approximate analysis.

These comments do not change my general opinion that the paper is very interesting, taking into account that colour measurements of NLC can be used for the particle size and altitude determination.

**Reply**: We are not completely sure if we understand this comment correctly. Yes, you are right, the reddening near the horizon depends on the aerosol loading of the lowest atmospheric layers. But our aim was not to simulate the evening sky accurately, but to test the influence of NLCs near the horizon. However, we included this point in Sect. 3.4: "It should be noted that the tropospheric aerosol loading is highly variable and the colours of the twilight sky may differ. But the main point here is the effect of NLCs on the reddish colour of the horizon.".

**Additional**:
Besides minor typing errors, we corrected the relative difference in L. 198 from 19% to 23%.

---

## Author Response (AR1)

Dear Dr. Paulino,

thank you! We uploaded all necessary documents.

Best regards,

Anna Lange